# Spontaneous shock waves in pulse-stimulated flocks of Quincke rollers

Bo Zhang [1,6] ✉, Andreas Glatz [1,2], Igor S. Aranson [3,4,5] & Alexey Snezhko [1] ✉

Active matter demonstrates complex spatiotemporal self-organization not accessible at equilibrium and the emergence of collective behavior. Fluids comprised of microscopic Quincke rollers represent a popular realization of synthetic active matter. Temporal activity modulations, realized by modulated external electric fields, represent an effective tool to expand the variety of accessible dynamic states in active ensembles. Here, we report on the emergence of shockwave patterns composed of coherently moving particles energized by a pulsed electric field. The shockwaves emerge spontaneously and move faster than the average particle speed. Combining experiments, theory, and simulations, we demonstrate that the shockwaves originate from intermittent spontaneous vortex cores due to a vortex meandering instability. They occur when the rollers' translational and rotational decoherence times, regulated by the electric pulse durations, become comparable. The phenomenon does not rely on the presence of confinement, and multiple shock waves continuously arise and vanish in the system.

Active matter encompasses a broad class of interacting self-propelled particles that transduce energy from the environment into mechanical motion[1–5]. With the increase in particle concentration, active matter exhibits a transition from individual to collective behavior manifested by various patterns of coherent locomotion: jets, bands, flocks, vortices[6–9]. This behavior was observed in many realizations of active matter, from macroscopic bird flocks, fish schools to microscopic bacterial suspensions, cytoskeletal extracts, and field-driven Janus particles, spinners and rollers[10–20].

Microscopic Quincke rollers are a popular realization of synthetic active matter. Quincke rollers—dielectric colloids suspended in a weak electrolyte and energized by a static (DC) electric field—utilize the electrohydrodynamic Quincke rotation phenomenon[21,22] and inject energy and angular momentum into the system at the microscopic level. In the presence of a solid surface, the Quincke rotation is transformed into a horizontal translation. Quincke rollers demonstrate a remarkable level of complex collective behaviors and self-organization ranging from the emergence of correlated flocks to the formation of global vortices, polar bends, and oscillating flows under confinement[7,23–25]. Temporal modulation of the activity of Quincke rollers via a pulsed electric field is an effective technique to control the persistence lengths and collective behavior of rollers[26–28]. By manipulating the duration $\tau_{\mathrm{on}}$ and intervals $\tau_{\mathrm{off}}$ between the pulses of the same polarity, a set of novel dynamic states, such as multiple localized vortices and lattices emerge[28]. The new patterns are often attributed to dynamic system memory and changing interparticle force balances at time scales comparable to the Maxwell-Wagner polarization relaxation time $\tau_{\mathrm{MW}} = (\epsilon_{\mathrm{p}} + 2\epsilon_{\mathrm{f}})/(\sigma_{\mathrm{p}} + 2\sigma_{\mathrm{f}})$, where $\epsilon_{\mathrm{p,f}}$ and $\sigma_{\mathrm{p,f}}$ are respective particle and fluid permittivities and conductivities[26,28].

The suspending media also plays a significant role in active ensembles dynamics[29–33]. In the case of Quincke rollers, the observed complex dynamics of rolling colloids is always accompanied by

[1]Materials Science Division, Argonne National Laboratory, Lemont, IL 60439, USA. [2]Department of Physics, Northern Illinois University, DeKalb, IL 60115, USA. [3]Department of Biomedical Engineering, Pennsylvania State University, University Park, PA 16802, USA. [4]Department of Chemistry, Pennsylvania State University, University Park, PA 16802, USA. [5]Department of Mathematics, Pennsylvania State University, University Park, PA 16802, USA. [6]Present address: Collaborative Innovation Center of Advanced Microstructures, National Laboratory of Solid State Microstructure, and Department of Physics, Nanjing University, Nanjing 210093, China. ✉e-mail: bz@nju.edu.cn; snezhko@anl.gov

electrohydrodynamic flows induced by the applied electric field powering the system. The strength of the flows grows with the amount of charge in the media, which in the case of the majority of Quincke experimental systems is regulated by the ionic surfactant AOT (aerosol dioctyl sulfosuccinate sodium) salt and the absorbed water content[34–36].

Here, we report on the emergence of spontaneous shockwaves that became accessible under temporal activity modulations in crowds of colloidal Quincke rollers with the increased strength of the electrohydrodynamic flows. In response to the increased media conductivity, the electrohydrodynamic flows are no longer negligible and promote intermittent rollers densifications (dynamic ripples) in the system at the uncorrelated gas state. The particle shockwaves continuously emerge and dissipate on the background of spontaneous density variations in the transition region between the gas and vortex states. The shockwaves originate in local high-density regions where rollers develop velocity correlations and spontaneously start to move collectively faster than the average particle speed in the ensemble. The dependent velocity distributions have also been observed in related magnetic roller systems[37]. We combine experiments and continuum computational modeling to demonstrate that the shock waves originate from the transient vortex cores due to vortex meandering instability and occur when the active rollers' translational and rotational decoherence times become comparable. Multiple shock waves continuously appear and vanish in the system. Our work highlights the crucial importance of the interaction timescales in the emergence of dynamic patterns under temporal modulation of the activity and suggests pathways to manipulate and enrich collective dynamics in active systems.

## Results

In our experiments, we use polystyrene spheres ($d = 4.8$ μm) dispersed in a weakly conductive liquid that are sandwiched between two ITO-coated glass slides and energized by a static (DC) electric field (see "Methods" for the details). Above a certain threshold value of the field strength, $E_c$, the particles start to spontaneously rotate due to the electrohydrodynamic Quincke rotation phenomenon[22] and turn into rollers exploring the bottom plate of the experimental cell. A typical velocity of the rollers under a static electric field $E = 3.2$ V/μm used in our experiment is 1.9 mm/s. The motion of the particles proceeds over the background of the electrohydrodynamic (EHD) flows between the electrodes. The EHD flows are always present in the energized system and scale with the amount of charge available in the liquid and the applied electric field. However, the effect of the EHD flows on the observed particles dynamics under typical field conditions for the Quincke rollers is often negligible. At high electric field, the EHD flows may become dominant and result in lifting off the particles and three-dimensional patterns[33]. To manipulate the electrohydrodynamic flows in our experiments, we increase the medium's conductivity by absorbing water to $\sigma = 7.8 \times 10^{-8}$ S/m.

The behavior of the rollers becomes significantly different if the activity of particles is modulated by the pulsed electric field[26,28]. Figure 1A demonstrates the dynamic phase diagram in our system under pulsed electric field excitations as a function of the pulse duration, $\tau_{on}$, and the interval between the pulses, $\tau_{off}$. The magnitude of the pulses was fixed at $E = 3.2$ V/μm. The system now exhibits a new striking collective response−spontaneous shockwaves−contentiously emerging at different locations of the ensemble, propagating and dissipating. Typical shockwave fronts are shown in Fig. 1B (see also Supplementary Movie 1). The new dynamic phase has not been previously observed at lower medium conductivity ($\sigma = 5.4 \times 10^{-8}$ S/m) under identical activity modulations[28] where only flocks, vortices, and lattices have been observed (Supplementary Fig. 1). Noticeably, the shock waves propagate on top of the spontaneous local particle number densifications, ripples, clearly visible in Supplementary

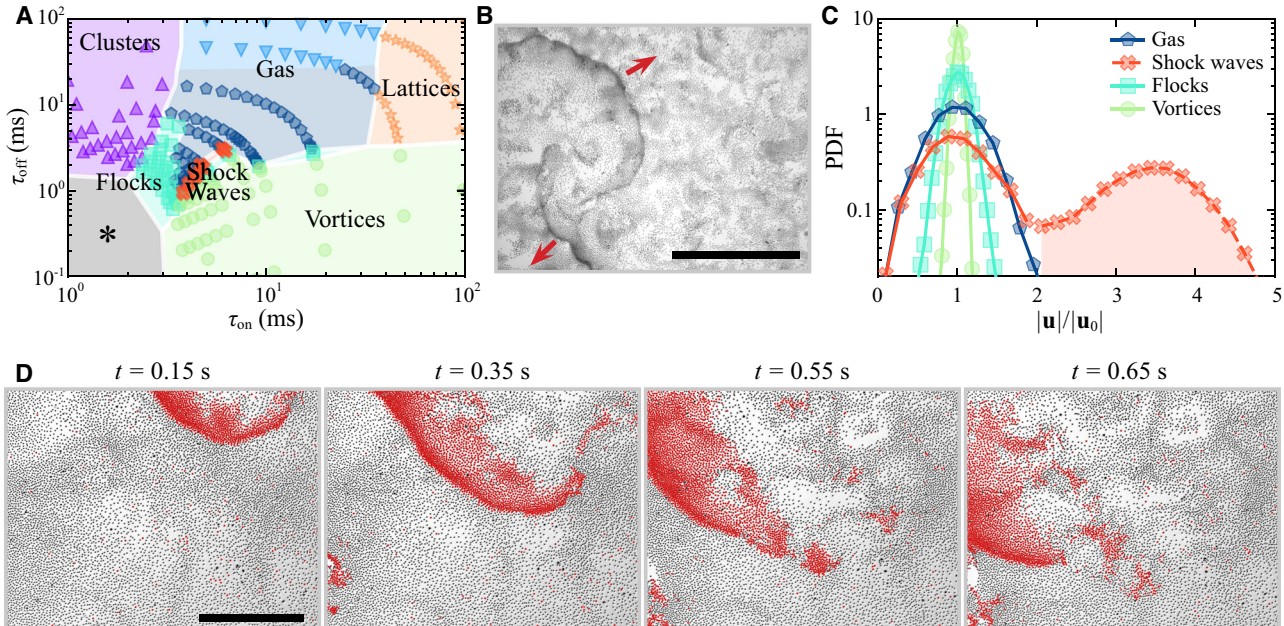

**Fig. 1 | Dynamic phases formed by Quincke rollers under a pulsed electric field.** **A** Phase diagram as a function of pulse duration $\tau_{on}$ and the pulse interval $\tau_{off}$. The phases in the gray area marked with * were reported previously in ref. 26. The average particle area fraction $\phi_0 = 0.11$. The conductivity of the medium $\sigma = 7.8 \times 10^{-8}$ S/m. The field magnitude $E = 3.2$ V/μm. See Supplementary Movies 1–5 and Supplementary Fig. 3 for more details on dynamic phases. **B** Experimental snapshot of multiple shock waves. The red arrows indicate the propagation directions of two major waves. The scale bar is 1 mm. **C** Representative probability distribution functions (PDFs) of the particle velocities for different dynamic phases: gas ($\tau_{on} = 4.0$ ms; $\tau_{off} = 2.7$ ms), shock waves ($\tau_{on} = 4.9$ ms; $\tau_{off} = 1.8$ ms), flocks ($\tau_{on} = 5.3$ ms; $\tau_{off} = 1.4$ ms) and vortices ($\tau_{on} = 6.6$ ms; $\tau_{off} = 0.1$ ms). The period of excitation $T = 6.7$ ms. The second peak of PDF for the shock waves regime is shaded by pink, indicating fast particles ($|\mathbf{u}|/|\mathbf{u}_0| > 2$) involved in the shock waves. $|\mathbf{u}_0|$ is the average particle speed in the first peak in the shock waves regime and the average speed of all particles in other phases. **D** Snapshots of a shock wave propagating from top right to bottom left. Particles with ($|\mathbf{u}|/|\mathbf{u}_0| > 2$) are colored in red. The scale bar is 0.5 mm. Source data are provided as a Source Data file.

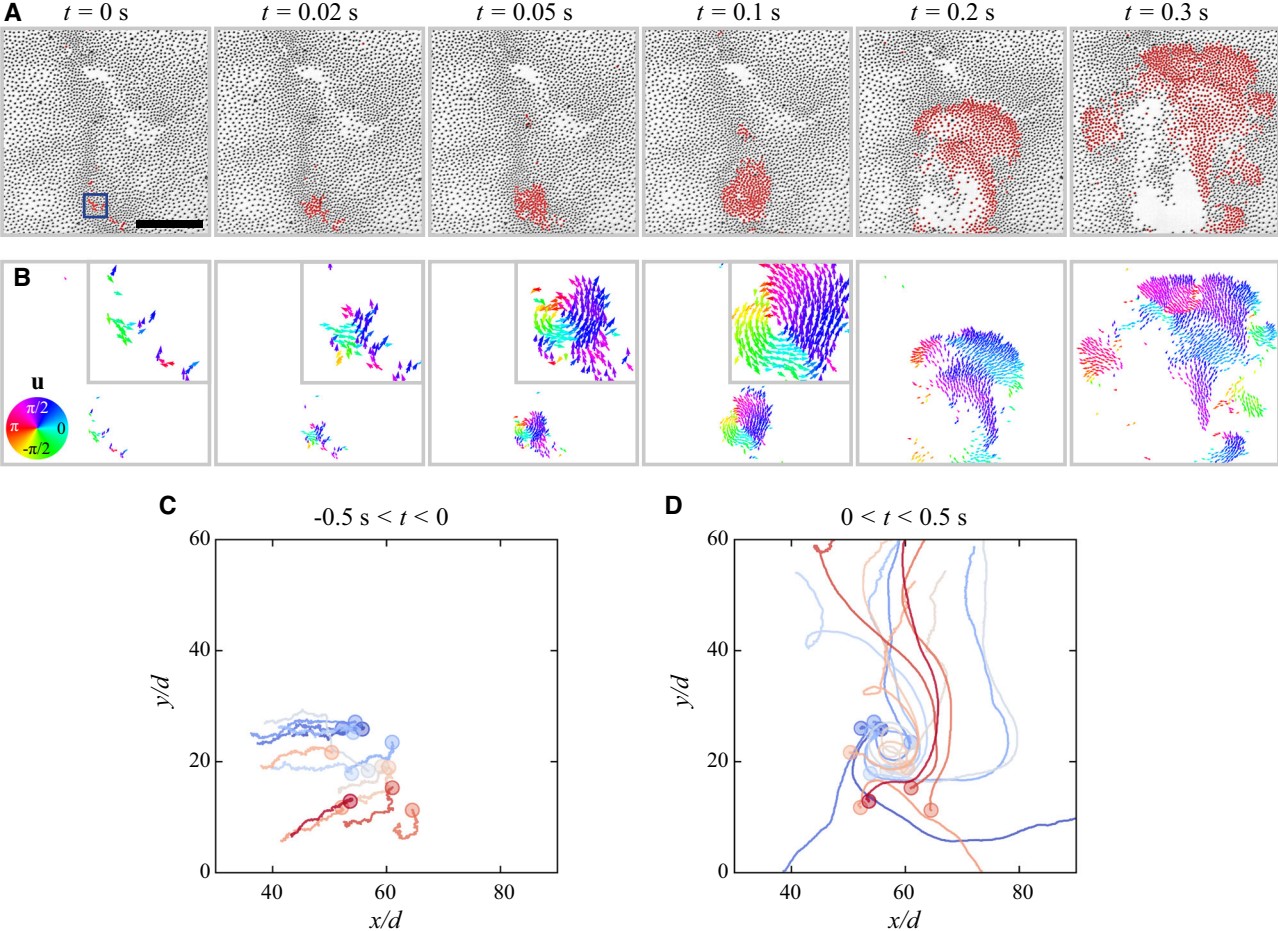

**Fig. 2 | Emergence of a shock wave. A** Snapshots illustrating the excitation of a shockwave. Fast particles ($|\mathbf{u}|/|\mathbf{u}_0| > 2$) are colored in red. The scale bar is 0.2 mm. **B** Velocity vectors of fast particles colored according to the velocity directions. Insets are zoomed-in region containing fast particles. A transient vortex is formed before transformation into a shockwave. **C**, **D** Trajectories of the particles inside of a square indicated in (**A**) before (**C**) and after (**D**) the shockwave excitation. Particle positions at $t = 0$ are marked with circles. $d$ is the particle diameter. Only 10% particle trajectories are shown for a better visualization.

Movies 1 and 2. The state of the system at exactly the same driving field conditions as the ripples state demonstrated but at lower media conductivity ($\sigma = 5.4 \times 10^{-8}$ S/m) is shown in Supplementary Movie 9. Instead of ripples, the system organizes in a vortical motion without significant densifications. In principle, the ripple-free gas state at low conductivity is also achievable but at different driving conditions, see Supplementary Movie 10. The formation of the ripples is driven by electrohydrodynamic flows between the conductive plates that, in the presence of the particles at the electrode distorting the local electric field, result in tangential fluid flows directed towards the particles giving rise to an effective interparticle attraction and local particle densifications[33,38,39]. Shockwaves are spontaneously excited from the local transient densification regions of ripples generating propagating spiral-like wavefronts that will eventually dissipate or be interrupted by another shockwave propagating in the system, see Fig. 1D and Supplementary Movie 3.

The build-up of the particle velocity correlations in the system resulting in collective phases is controlled by the activity modulations. A particle activity and retained polarization memory increase with increasing $\tau_{on}$ and/or shortening $\tau_{off}$ (see Supplementary Fig. 2). The probability distribution function (PDF) of the particle velocities in different dynamic phases of the system are shown in Fig. 1C. The regime of shockwaves has two distinctive peaks in its distribution corresponding to the particles in a gas phase performing uncorrelated motion (low-velocity peak) and fast particles involved in the intermittent shockwaves characterized by a short-lived correlated motion

of the particles (high-velocity peak). The short-lived correlations between the particles in a shockwave are promoted by the particle densifications of the ripples driven by the electrohydrodynamic flows. Those correlations decay as the interparticle distances increase with the wave propagation, resulting in the eventual dissipation of the wavefront. The shock waves appear at the narrow transition region before the system switches to a vortex phase.

The excitation process of a shockwave is illustrated in Fig. 2 and Supplementary Movie 6. Rollers first slowly accumulate to dense vertices of ripples via constrained random walks under electrohydrodynamic flows (Fig. 2C). Accidentally, rollers gain high velocities ($|\mathbf{u}|/|\mathbf{u}_0| > 2$) and form small dynamic clusters with particle velocities aligned ($t = 0$). Most dynamic clusters dissipate over time or explode as ripples while a small cluster fraction merges ($t = 0.02$ s), grows ($t = 0.05$ s), and eventually forms a vortex ($t = 0.1$ s). Due to the vortex's meandering instability in an unconfined environment, the unstable vortex quickly breaks into a spiral shock wave ($t = 0.2$ s) which propagates ($t = 0.3$ s) and eventually dissipates. The excitation process is also presented by typical particle trajectories, which show the circular motion of the initial transient vortex and spiral trajectories of the final shock wave (Fig. 2D).

Rollers accelerate when a shockwave propagates through and forms a densified region associated with higher particle velocities. The shape of the shockwave is shown in the corresponding microscopy image, the particle density, and the particle velocity map in Fig. 3A–C. The shockwave bulges in the propagation direction, causing typical

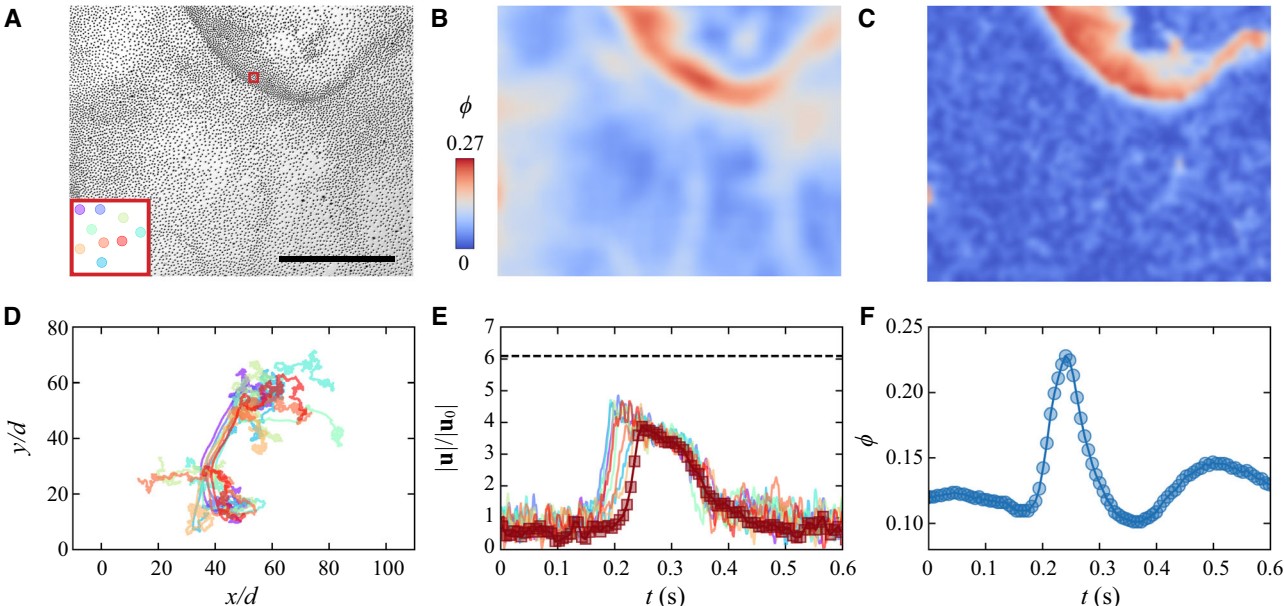

**Fig. 3 | Shockwave propagation. A**–**C** Microscopy image, particle density, and particle velocity maps of a propagating shock wave. The inset visualizes 9 particles in the red square ($5d \times 5d$) at $t = 0.25$ s. The scale bar is 0.5 mm. **D** Trajectories of particles in the red square are shown in (**A**). The trajectories' color code is the same as shown in (**A**) inset. The smooth and aligned middle parts of the trajectories indicate the passing of a shock wave, while the rest of the trajectories demonstrate uncorrelated random motions. **E** Corresponding speed evolution of particles (colored lines) shown in (**D**) and average velocity of particles in the fixed red square (dark red line with square symbols). The horizontal dashed line indicates the speed of the shock wave $u_{wave} = 2.1$ mm/s. The baseline of average velocity in the square is lower than $|\mathbf{u}_0|$ due to the average of velocities rather than speeds. See details about the calculation of the wave speed in Supplementary Fig. 4. **F** Variations of the particle local area fraction in the red square in the process of the shock wave passage. Source data are provided as a Source Data file.

asymmetric density and velocity profiles with sharply curved wavefronts and relatively shallow tails. To better understand the effect of the shockwave on individual rollers, we track a few selected particles and measure their instantaneous positions and velocities when a shock wave passes through (see Fig. 3D–E). Rollers first perform Brownian-like movements, showing random trajectories and slow speeds. When the shock wave arrives, particles move smoothly and collectively for about $50d$ before returning to their random motion (Fig. 3D). The distance for collective displacement is comparable to the width of the shockwave indicated by the particle density map or velocity map. During this process, the particle speeds dramatically increase to about 5 times and then decay relatively slowly to the original level, see Fig. 3E.

Besides tracking the particles, the influence of shock waves is also monitored by the evolution of the average velocity (dark red line with square symbols in Fig. 3E) and local particle area fraction (Fig. 3F) at a fixed position indicated by a red square in Fig. 3A. The shape of the average velocity evolution is similar to those of individual particles (colored lines) with an exception of a slightly delayed increase due to different objects of measurements. When the shock wave arrives at the selected position ($t \approx 0.2$ s), the average velocity increases dramatically, accompanied by an abrupt increase in local particle density. The wave nature is confirmed by the fact that the speed of the shock waves ($u_{wave} = 2.1$ mm/s) is about 40% higher than the peak particle speed ($u = 1.5$ mm/s). This makes the shock waves very different from other traveling density bands observed in many active matter systems where the particle velocity is the same as the band front velocity[7,8,40].

While the phenomenology seems somewhat similar to the activity waves reported recently in ref. 36, there are several crucial differences between our shockwaves and the activity waves observed in populations of subcritical Quincke rollers. Firstly, the shock waves of rollers are driven by a pulsed electric field with a field amplitude higher than the critical field strength $E_q$. In contrast, activity waves are observed for a constant electric field slightly lower than $E_q$. Therefore, rollers in shock waves perform steady rotations and induce hydrodynamic flows

during $\tau_{on}$. In contrast, in activity waves, the transient motion of particles is triggered by repulsion from nearest neighbors due to the Quincke instability. Secondly, the wave propagation mechanisms are also different. Rollers in shock waves interact via electrostatic and hydrodynamic interactions, while electrostatic interactions and hard-core collisions are dominant for the particle motion in activity waves. Due to long-range interactions via hydrodynamic flows, shock waves can be excited in relative dilute systems ($\phi \sim 0.1$), while activity waves are only observed at very high-density systems ($\phi \sim 0.4$) to trigger a domino-like effect due to hard-core collisions.

## Modeling and simulations

To study and understand the dynamical behavior of the system in response to variations of the $\tau_{on}$ and $\tau_{off}$ of the excitation electric field in the experiments, we investigate the behavior of the roller system using a continuum model and perform computational studies. Within the model, variations of the translational and rotational diffusion constants are directly affected by the activity modulation procedure. See the "Methods" section for details of the computational model. We study the steady-state behavior of the system depending on the parameter $\delta = \tilde{D}/\tilde{D}_r$, where the tilde denotes dimensionless parameters.

The particle density becomes homogeneous for smaller $\delta \lesssim 0.5$, which we associate with the gas states, whereas for large $\delta \gtrsim 1.16$, the system develops a stable global vortex. We find shockwave states for the region $\delta \in [0.6, 1.16]$, as shown by the snapshots of the steady-states for different $\delta$ values in Fig. 4A–D. The shockwave state becomes fully developed with multiple waves traversing the system at $\delta$ approaching 1, corresponding to the state where the roller's translational and rotational decoherence times become comparable. The dynamics of the shockwaves, as obtained in the simulations, is also shown in Supplementary Movies 7 and 8.

Since we are particularly interested in the shockwave regime, we performed a detailed analysis of this state in analogy to the experiments. Figure 4E, G shows the evolution of the particle density in the

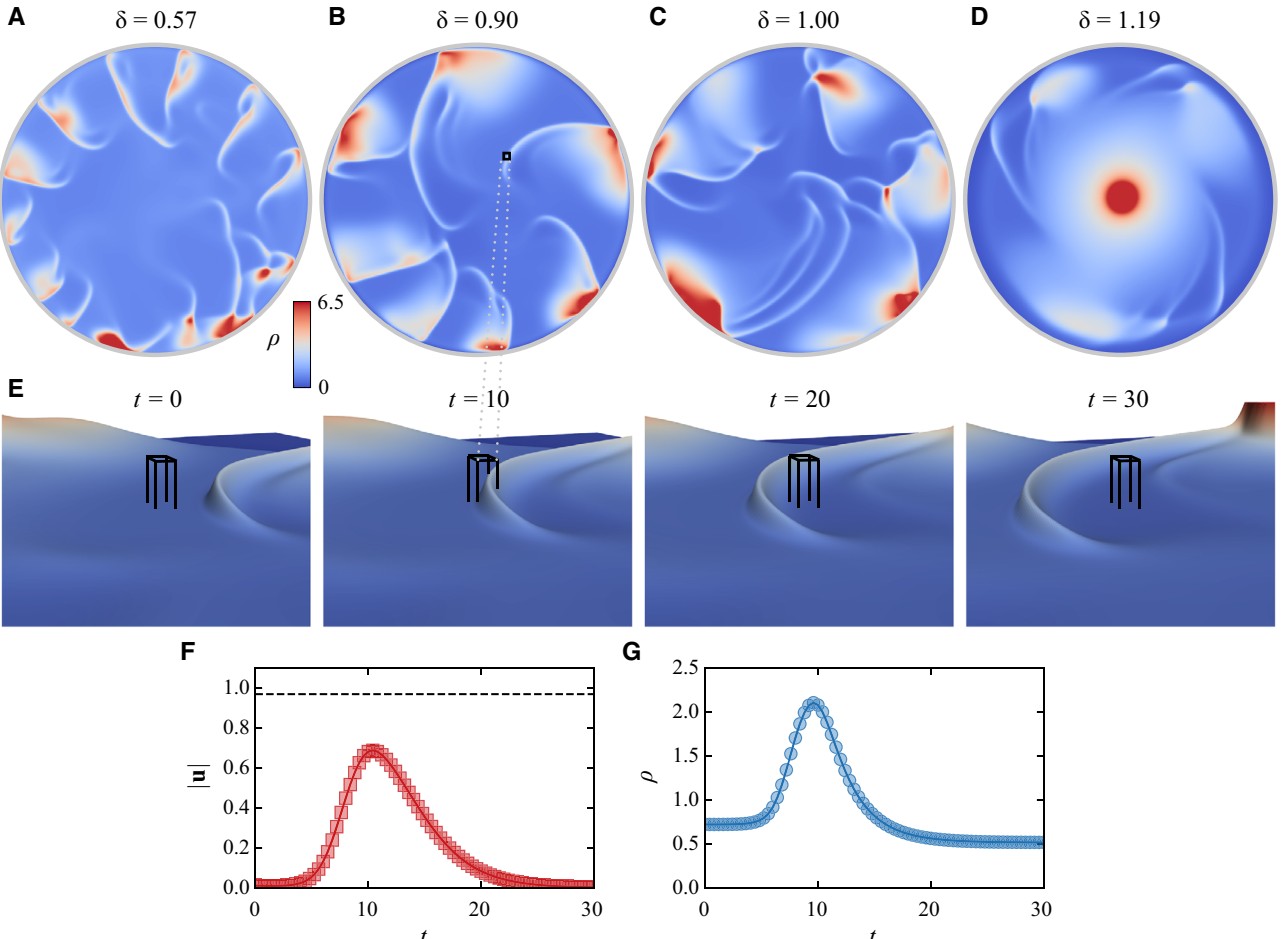

**Fig. 4 | Simulation of Quincke rollers confined in a large circular region using the continuum model. A–D** Snapshots of the steady-state particle density $\rho$ for different ratios $\delta = D/D_r$. The system shows a transition from a homogeneous state at low $\delta$ over to the state with the spontaneous shock waves (**A–C**) when the roller's translational and rotational decoherence times become comparable and further to a single vortex state for $\delta \geq 1.16$ (**D**). **E** Close-up of a shock wave traveling through a small analysis region (outlined) as 3D density isosurface. See also Supplementary Movies 7 and 8. **F, G** Time evolution of the particle speed $|\mathbf{u}|$ (**F**) and particle density $\rho$ (**G**) in the process of a shockwave passage through the analysis region shown in (**E**). At $t \sim 10$, the shockwave passes through the region resulting in a maximum speed of 0.68, while the wave travels at a speed of 0.97 (horizontal dashed line), i.e., 42% faster than the local particle speed. Source data are provided as a Source Data file.

selected region of interest. In this analysis region (outlined), we calculate the particle speed $|u|$ as a function of time while a shock wave travels through it. A detailed animation of this process can be found in the Supplementary Movie 8. For the particular shock wave shown in Fig. 4F, the wave passes through the analysis region at time $t \sim 10$ with a maximum local particle speed $|u|$ of $u_{max} = 0.68$ (Fig. 4G), while the wave travels at speed $u_{wave} = 0.97$ (dashed black line), i.e., 42% faster than the local particle speed. The analysis was performed for several shock waves with comparable ratios of $u_{wave}/u_{max}$, which agrees with the experimental observation.

Analysis of the shockwave excitation in the simulations reveals local instabilities that lead to non-zero vorticity, producing localized densification of particles forming short-lived vortices. The emission of radial shockwaves then dissolves these whirling, denser spots. Local densifications leading to the emergence of shockwave fronts also occur when flocks collide with each other or with the boundaries of the confinement potential creating intermittent density hot spots in the density map.

## Discussion

We have demonstrated that active ensembles of Quincke rollers with enhanced role of electrohydrodynamic flows exhibit the onset of spontaneous shockwaves that became accessible under temporal

modulations of activity by a pulsed electric field. The shock waves continuously emerge, propagate, and dissipate at different locations of the ensemble. The electrohydrodynamic flows are no longer negligible and promote intermittent rollers densifications (dynamic ripples) in the system. We have shown that shockwaves emerge at high-density regions and, like shockwaves in gases, propagate at a speed exceeding the average particle speed. These emergent waves originate from the transient vortex cores due to vortex meandering instability. The computational modeling sheds light on the origin of the observed shockwaves and reveals that this unconventional dynamic state becomes accessible when the translational and rotational decoherence times are comparable. The presented computational model does not consider the 3D electrohydrodynamic flows[33]. The current version of the model operates with the hydrodynamic flows in the shallow water approximation, i.e., quasi-two-dimensional geometry. The effect of the electrohydrodynamic flows can be further included via an additional vertical fluid velocity component similar to that in ref. 41. The extended model will likely provide a better agreement with the experimental observations.

In the context of shock waves, the Burgers equation is often used to describe their formation due to the competition between the viscosity and the convective nonlinearity $\mathbf{v}\nabla\mathbf{v}$. Since our equations contain the convective nonlinear terms, it is reasonable to assume, at least,

at the qualitative level, some resemblance of the shock wave formation mechanism as in the Burgers equation. The main difference, however, is that the shock waves are not the steady-state solutions in the Burgers equation. On the contrary, due to energy injection in our system, the shock waves propagate without change of shape. It would also be interesting to compare our results to a Burgers' equation approach akin ref. 42, since the nonviscous Burgers' equation gives rise to discontinuities resulting in shockwaves. Our findings highlight the importance of the interplay between transient system memory, manipulated by a pulsed field, and electrohydrodynamic flows in accessing unconventional dynamic phases that are not accessible under a continuous energy input. The results suggest new approaches for controlling and manipulating active colloidal materials at the microscale.

## Methods

### Experimental details

In our experiments, spherical polystyrene particles ($d = 4.8$ μm) are suspended in 0.15 mol/L AOT/hexadecane solution and injected into an open cell constructed by two parallel ITO-coated glass slides and spacers with a typical thickness of 95 μm. The electric field is supplied by a function generator (Agilent 33210A, Agilent Technologies) and a power amplifier (BOP 1000M, Kepco Inc.). The water content of the host fluid is controlled by the relative humidity (RH) of the closed environment and monitored by real-time conductivity measurements. The pulsed field amplitude was set to $E = 3.2$ V/μm. The sample cell is observed under a microscope with a 4× microscope objective. Videos are recorded by a fast-speed camera (IL 5, Fastec Imaging) at 1057 frames per second (FPS). Particle tracking velocimetry (PTV) and further data analysis are carried out with custom codes in Python and Trackpy[43].

### Computational approach

Here we employ a continuum model for Quincke rollers. The approach is adapted from our previous work[44,45], which we developed for magnetic roller systems. Within this model, the particles are described by a coarse-grained particle density field $\rho(\mathbf{r})$ and their velocity-field $\mathbf{u}(\mathbf{r})$. The fluid is described by the fluid velocity $\mathbf{v}(\mathbf{r})$ and fluid height profile $h(\mathbf{r})$. The host fluid is described by the depth-averaged in-plane velocity $\bar{\mathbf{v}} = \bar{\mathbf{v}}(x,y,t)$, i.e., we use a shallow water approximation here, and the depth of the solvent $h = h(x, y, t)$. The gravity role is auxiliary and used for simplification purpose. The term $-g\nabla h$ in Eq. (2) is due to the shallow-wave approximation and is required to satisfy the fluid incompressibility condition, Eq. (3). It is possible to set $h = const$, and, correspondingly, $\nabla \bar{\mathbf{v}} = 0$. However, it would result in a more computationally challenging algorithm without affecting the observed behavior. The particle dynamics is described by a Ginzburg-Landau-like equation for $\mathbf{u}$:

$$\partial_t \mathbf{u} = \alpha\mathbf{u} - \beta|\mathbf{u}|^2\mathbf{u} + D\nabla^2\mathbf{u} + \frac{1}{\rho m_0}\nabla\cdot\Pi + \gamma\rho\bar{\mathbf{v}} + \boldsymbol{\Omega}\times\mathbf{u}. \quad (1)$$

where $m_0$ is the mass of a roller, $D = u_0^2\tau_{\text{dif}}/4$ is the translational diffusion coefficient of the particles. The Ginzburg-Landau parameters are determined by the mean collision and diffusion times, $\tau_{\text{col}} = (2\rho a_0 u_0)^{-1}$ and $\tau_{\text{dif}}$, respectively, as $\alpha = \tau_{\text{col}}^{-1} - \tau_{\text{dif}}^{-1}$ and $\beta = (u_0^2\tau_{\text{col}})^{-1}$. The former is described in dimensionless units as $\tilde{\alpha} = \eta u_0\tilde{\rho} - \tilde{D}_r$, where $\tilde{D}_r$ is the dimensionless rotational diffusion constant, $\sim\tau_{\text{dif}}^{-1}, \eta$ a numerical constant, and $\tilde{\rho}$ the dimensionless particle density. The last two terms characterize the coupling between active rollers and a passive host fluid (solvent), where the $\gamma = \frac{3}{4}\frac{a_0}{h}\tau_{\text{col}}^{-1}$ term results from the over-damped roller dynamics, and the last term describes the rotation of rollers in a hydrodynamic flow with vorticity $\boldsymbol{\Omega} = \frac{1}{2}\nabla\times\bar{\mathbf{v}}$[46]. For the Quincke roller system, the stress tensor $\Pi$ takes the form $\Pi = \frac{3}{64a_0}(\boldsymbol{p}\otimes\boldsymbol{p} - \frac{3}{2}p^2\mathsf{I}) - P\mathsf{I}$, where $\boldsymbol{p}(\mathbf{r})$ is the polarization

field, $\mathsf{I}$ is the identity tensor, and $P$ the pressure, which phenomenologically accounts for the finite size of colloids. The latter results in a term $-Q(\rho)\nabla\rho$ in Eq. (1), where $Q(\rho)$ takes into account hard-core repulsion at high densities (i.e., when two particle overlap) and attraction for intermediate densities, which accounts for polarization effects being linear in $\rho$, and a small repulsion at very low densities[47]. The polarization field $\boldsymbol{p}$ is itself a dynamic quantity, similar to $\boldsymbol{u}$, and therefore described by a related Ginzburg-Landau equation with Landau-Lifshitz-like term aligning $\boldsymbol{p}$ and $\boldsymbol{u}$, see ref. 44.

In weakly-conducting fluids, the interactions between dipoles scale as separation distance in power four[48]. In the so-called leaky-dielectric model[48,49], the fluid's conductivity decreases the dipole strength. The dipolar interactions are small compared to the hydrodynamic interactions that decay much slower[41]. Thus, unlike in a magnetic system, the electrostatic dipolar interactions between Quincke rollers become negligible compared to the hydrodynamic interactions and we can neglect the dipolar contribution to the stress tensor $\Pi$ in Eq. (1).

For the dynamics of suspending fluid, we use the two-dimensional depth-averaged Navier-Stokes equation (shallow water approximation)[50]

$$\partial_t\bar{\mathbf{v}} + (\bar{\mathbf{v}}\cdot\nabla)\bar{\mathbf{v}} = -g\nabla h + \nu\nabla^2\bar{\mathbf{v}} - 3\frac{\nu}{h^2}\bar{\mathbf{v}} + 3\pi\rho a_0^2\frac{\nu}{h^2}\mathbf{u}, \quad (2)$$

where $g$ is the gravitational acceleration and $\nu$ the kinematic viscosity. The last two terms on the RHS originate from the no-slip condition at the rollers-solvent interface. $u_0$ also determines the scale of the fluid velocity.

Equations (1) and (2) have to be solved together with the continuity equations for the

$$\partial_t\rho + \nabla\cdot\rho\mathbf{u} = 0, \quad (3)$$

$$\partial_t h + \nabla\cdot h\bar{\mathbf{v}} = 0. \quad (4)$$

All equations are integrated using quasi-spectral split-step methods, which calculate all second-order spatial derivatives in Fourier space. Technically, the solver is implemented on the general-purpose graphics processing units (GPU) using complex fast-Fourier-transforms (FFT; here the cuFFT implementation) for the $x$ and $y$ components of $\mathbf{u}, \bar{\mathbf{v}}$, and the combined $(h, \rho)$ vector. Compared to general-purpose CPU finite-element solvers, this method allows for an integration speed-up of 3 to 4 orders of magnitude and naturally uses periodic boundary conditions due to the FFTs.

**Simulations parameters.** Using $\tau_{\text{dif}}$ as unit of time and $u_0\tau_{\text{dif}}$ as unit of length the above equations are rewritten in dimensionless units. The roller density is normalized by the mean value $\bar{\rho} = \nu_p/(\pi a_0^2)$, where $\nu_p$ is the surface fraction of the particles. A dimensionless parameter $\rho_0 < 1$ determines then the average density in the system.

The units are defined by their experimental values, which set the following dimensionless parameter ranges for the simulations

$$D \approx 1$$
$$D_r \sim 0.8 - 2.0$$
$$\rho_0 \approx 0.4$$

To solve the above equations numerically, a time unit is discretized in 250 steps, and the system is partitioned spatially on a regular, square mesh with up to 2048 × 2048 grid points. Additionally, the equation for the particle velocity has an additional circular confinement force, which is zero inside the circular region of diameter comparable to linear system size. This confinement is used to mimic the experimental geometry and to avoid an overall transversal mode

due to the needed periodic boundary conditions for the FFT used to solve the equations of motion. The equations are then integrated for up to $10^7$ time steps, corresponding to about 5 min experimental time.

## Data availability
All data that support the findings of this study are provided in this paper and the Supplementary Information. Source data are provided with this paper.

## Code availability
Custom codes used for numerical modeling are available at github.com/activematerials/Shockwave_continuum.

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

## Acknowledgements

The research at Argonne National Laboratory was supported by the U.S. Department of Energy, Office of Science, Basic Energy Sciences, Materials Sciences and Engineering Division. Use of the Center for Nanoscale Materials, an Office of Science user facility, was supported by the U.S. Department of Energy, Office of Science, Basic Energy Sciences, under Contract No. DE-AC02-06CH11357. The research of ISA was supported by the U.S. Department of Energy, Office of Science, Basic Energy Sciences, under Award no. DE-SC0020964.

## Author contributions

A.S. and B.Z. conceived the research, B.Z. performed the experiments, A.G. and I.S.A. conducted the numerical simulations and formulation of the model. All authors analyzed the data and wrote the manuscript.

## Competing interests

The authors declare no competing interests.
