## [Peer Review File · Nature Communications]

REVIEWER COMMENTS

Reviewer #1 (Remarks to the Author):

This manuscript reports the formation of shock waves in collections of Quincke rollers subject to pulsed electric fields. Shock waves are characterized by propagating regions of high particle density and velocity. Such behavior is found within a narrow region of operating conditions where the “on” and “off” times of the pulsed field are comparable to rotation frequency of the rollers. Shock waves emerge spontaneously from transient vortices that form in local regions of high particle density. Experimental observations are reproduced qualitatively by a hydrodynamic model that describes the evolution and interaction of particle and fluid velocity fields. The model suggests that spontaneous shock waves can occur when the translational and rotational diffusivities are comparable.

The shock wave is similar to that formed by magnetic rollers (ref. 39), where hydrodynamic interactions lead to a density dependent rolling speed. Here, however, the speed of the shock wave exceeds that of the particles within. The shock wave is also distinct from activity waves reported for subcritical Quincke rollers (ref. 36).

The experimental characterization of this novel phenomenon is thorough and clearly described. The simulation results provide useful insights into the likely mechanism. The use of pulsed fields to access emergent behaviors in the Quincke system suggests opportunities for further discovery and control of these collective effects.

I recommend publication in Nature Communications. I would appreciate it if the Authors could respond to the following questions / comments in their manuscript.

(1) Why does the small change in fluid conductivity have such an effect? The claim that the fluid conductivity manipulates “the strength of electrohydrodynamic flows” seems speculative. Can the Authors elaborate on this effect?

(2) Electrohydrodynamic flows are also invoked to explain the “dynamic ripples” in the gas state. Can the Authors provide data showing the ripple-free gas state for the lower conductivity conditions?

(3) The description of the simulations provided in the methods appear insufficient to reproduce the numerical results. Can the Authors provide additional details in the supporting information? In particular, the different contributions to the stress tensor are discussed qualitatively but not explicitly stated.

(4) Page 16. "Quincke rollers are exponentially screened." The validity of this claim is questionable. In the leaky-dielectric model, the field-induced dipole moment is not screened despite the presence of the conductive medium. The Debye length describes charge separation due to thermal motion, which is negligible compared to field-induced charge separation at the high field strengths used herein. Electric interactions may indeed be secondary as compared to hydrodynamic interactions; however, the discussion of their relative magnitude should be revised.

Reviewer #2 (Remarks to the Author):

Very interesting paper on the formation of the shock waves in the suspension of the Quincke rollers. Before the publication I suggest to the authors consider some optional remarks which to my opinion would give some additional insights in the physics behind the observed phenomena.

1. The observed phenomena to my opinion resemble the shock waves described by the Burgers equation. In the suspension of the Quincke rollers such phenomena are possible due to the dependence of the velocity of rollers on the concentration of particles (M.Belovs, A.Cebers, PRE, 89, 052310 (2014)), Dependence of the velocity of rolling on the concentration of particles is studied experimentally (G.Junot et al, Soft Matter,17,8605 (2021)). I would like to see some comments on that.

2. Polarization equation of Quincke particles predicts orientation of particle rotation plane perpendicularly to the vorticity and the formation of the vortex of rolling particles due to the negative viscosity effect (M.Belovs, A.Cebers, Phys.Rev.Fluids,5,013701 (2020)). Interesting that it takes place even at electric field strength less than the threshold field when the spontaneous rotation of individual particles arises. Since, as far as I see, Eq.(2) does not contain electric field it is not clear if this phenomenon is captured by the model (maybe $u_{\{0\}}$ depends on electric field?). It seems even more important since the vortex formation is mentioned as one of the issues playing role in the formation of the shock waves.

3.Minor. What is the role of the gravity, accounted for in Eq.(2), in the observed phenomena.

Reply to Reviewer 1

We thank the Reviewer for a careful reading of our manuscript and comments. We are glad that the Reviewer finds our work novel and finds it appropriate for publication in Nature Communications. The Reviewer made a number of comments and suggestions, which we fully addressed in the revised manuscript. Below are point-by-point replies to the Reviewer's comments:

- 1) *Why does the small change in fluid conductivity have such an effect? The claim that the fluid conductivity manipulates “the strength of electrohydrodynamic flows” seems speculative. Can the Authors elaborate on this effect?*

Indeed, this point needs some clarification. By that statement, we just meant that the electrohydrodynamic flows between the electrodes scale with the amount of charge available in the liquid. These flows are always present in the system (however, their effect on observed particle dynamics is often negligible). The densification effects of these flows on particles are well documented previously (we cite a few references [Refs 37, 38] in the paper), however, one needs to reach certain densifications between the particles herded by these flows to promote sudden changes in the particle behavior (such as shockwaves). We clarified our statement and extended our discussion in the revised manuscript.

- 2) *Electrohydrodynamic flows are also invoked to explain the “dynamic ripples” in the gas state. Can the Authors provide data showing the ripple-free gas state for the lower conductivity conditions?*

Following the Reviewer's suggestion we now provide additional experimental videos demonstrating the behavior of the system at similar driving conditions as ripples but at lower conductivity of the media. The supplemental movie S9 demonstrates the state of the system at exactly the same driving field conditions ($E=3.2\text{V}/\mu\text{m}$; $\tau_{\text{on}}=4.0\text{ms}$; $\tau_{\text{off}}=2.7\text{ms}$) as the ripples state in the movie S1 but at lower media conductivity. Instead of ripples, the system organizes in a vortical motion under such conditions, nevertheless without the ripples (so the dynamic state is ripple-free).

In principle, the ripple-free gas state at low conductivity is also achievable but at different driving conditions. We provide also the video of the ripple-free gas state as the supplementary movie S10.

- 2) *The description of the simulations provided in the methods appear insufficient to reproduce the numerical results. Can the Authors provide additional details in the supporting information? In particular, the different contributions to the stress tensor are discussed qualitatively but not explicitly stated.*

We extended the description of the simulation methods (Methods section) to include the explicit expression for the stress tensor contributions.

- 4) *Page 16. “Quincke rollers are exponentially screened.” The validity of this claim is questionable. In the leaky-dielectric model, the field-induced dipole moment is not screened despite*

the presence of the conductive medium. The Debye length describes charge separation due to thermal motion, which is negligible compared to field-induced charge separation at the high field strengths used herein. Electric interactions may indeed be secondary as compared to hydrodynamic interactions; however, the discussion of their relative magnitude should be revised.

We thank the Reviewer for bringing this point to our attention that requires a clarification. In the leaky-dielectric model, the interaction between point dipoles scales like one over separate distance in power 4, $1/r^4$, see, e.g., Petia Vlahovska, *Electrohydrodynamics of Drops and Vesicles*, Annual Reviews of Fluid Mechanics, 2019, v51, 305). The main approximation is that in the leaky dielectric model, all charge is ascribed to the interface, and the dipole is given by the sum of the instantaneous (dielectric) and free charge (Maxwell-Wagner) polarization. The leaky dielectric model does not consider space charge and ions that could result in Debye-like screening; the effective “screening” is implicitly taken into account by the modified dipole strength. These interactions are much smaller than the hydrodynamic interactions between the particles, which could be as large as $1/r$ in the plane containing the particles, see. e.g., I. S. Aranson and M. V. Sapozhnikov, *Theory of Pattern Formation of Metallic Microparticles in Poorly Conducting Liquids*, Phys. Rev. Lett. 92, 234301 (2004). These interactions can be attractive for the large applied field values and could result in particle clustering. However, when these effects dominate, we always apply an electric field below the values.

To clarify this point, we modified the text. Now it reads: In weakly-conducting fluids, the interactions between dipoles scale as separation distance in power four. In the so-called leaky-dielectric model, the fluid’s conductivity decreases the dipole strength. The dipolar interactions are small compared to the hydrodynamic interactions that decay much slower. Thus, unlike in a magnetic system, the electrostatic dipolar interactions between Quincke rollers become negligible compared to the hydrodynamic interactions. Therefore, we can neglect the dipolar contribution to the stress tensor Π in Eq.1.”

Reply to Reviewer 2

We thank the Reviewer for the careful reading of our work and comments aimed at improving the clarity of our presentation. We are pleased by the positive evaluation of our work by the Reviewer. The Reviewer made a number of comments and suggestions which we fully addressed in the revised manuscript. Below are point-by-point replies to the Reviewer's comments:

1) *The observed phenomena to my opinion resemble the shock waves described by the Burgers equation. In the suspension of the Quincke rollers such phenomena are possible due to the dependence of the velocity of rollers on the concentration of particles (M.Belovs, A.Cebers, PRE, 89, 052310 (2014)), Dependence of the velocity of rolling on the concentration of particles is studied experimentally (G.Junot et al, Soft Matter,17,8605 (2021)). I would like to see some comments on that.*

We thank the reviewer for pointing out the Burgers approach for describing Quincke systems. The Burgers equation is often used to describe the formation of shock waves due to the competition between the viscosity and the convective nonlinearity $v \nabla v$. Since our equations contain the convective nonlinear terms, it is reasonable to assume, at least, at the qualitative level, some resemblance of the shock wave formation mechanism as in the Burgers equation. The main difference, however, is that the shock waves are not the steady-state solutions in the Burgers

equation. On the contrary, due to energy injection in our system, the shock waves propagate without change of shape.

Regarding the cited papers. In the PRE paper, the authors first derived kinetic equations for the distribution functions of the rotating particles, which are then combined with the Blake solution for the flow to describe an ensemble of Quincke rollers. This also results in a continuum model for the distribution function of the orientation of the particle rotation planes. This is used to derive a stationary spatially uniform solution and first-order deviations from it. It would, indeed, be interesting if a comprehensive numerical solution could give similar results to ours as the non-viscous Burgers' equation can give rise to discontinuities. We added a brief discussion and cited this paper and the Soft Matter reference where a collective speed increase with particle density in magnetic roller systems was observed. This is related to the observed densification effects resulting in high velocities and finally shockwaves.

We incorporated corresponding discussions in the text of the revised manuscript.

2) Polarization equation of Quincke particles predicts orientation of particle rotation plane perpendicularly to the vorticity and the formation of the vortex of rolling particles due to the negative viscosity effect (M.Belovs, A.Cebers, Phys.Rev.Fluids,5,013701 (2020)). Interesting that it takes place even at electric field strength less than the threshold field when the spontaneous rotation of individual particles arises. Since, as far as I see, Eq.(2) does not contain electric field it is not clear if this phenomenon is captured by the model (maybe $u_{\{0\}}$ depends on electric field?). It seems even more important since the vortex formation is mentioned as one of the issues playing role in the formation of the shock waves.

In our case the electric field is always above the threshold for the steady Quincke rotation when the spontaneous rotation of individual particles arises. We fix the magnitude of the field and vary the pulse lengths to go through the phases. We use pulsed field (instead of a continuous one) to modulate the particle motion. The model was developed with the assumption of the particle rotations. Our model cannot describe field strengths below the threshold field.

3) Minor. What is the role of the gravity, accounted for in Eq.(2), in the observed phenomena.

The gravity role is auxiliary and used for simplification purpose. The term ∇h is due to the shallow-wave approximation and is required to satisfy the fluid incompressibility condition, Eq. 4. It is possible to set $h = \text{const}$, and, correspondingly, $\nabla v = 0$. However, it would result in a more computationally challenging algorithm without changing the observed behavior.

Here the gravitational term describes the effect of surface waves, described by the fluid depth field h . We added the description of h and a sentence explaining this gravitational term, which essentially describes the acceleration or deceleration of the fluid velocity due to the presence of surface waves. However, in our simulations, we set the average depth to two particle diameters. The resulting gradient term of the depth field has only a minor effect on the particle dynamics and therefore on the formation of shockwaves.

REVIEWERS' COMMENTS

Reviewer #1 (Remarks to the Author):

The Authors have addressed by previous questions with suitable revisions of the main text and the supporting information.

Reviewer #2 (Remarks to the Author):

Authors gave satisfactory responses to the issues noted in my first report so I can recommend now the paper for publication.

REVIEWERS' COMMENTS

Reviewer #1 (Remarks to the Author):

The Authors have addressed by previous questions with suitable revisions of the main text and the supporting information.

Reviewer #2 (Remarks to the Author):

Authors gave satisfactory responses to the issues noted in my first report so I can recommend now the paper for publication.

Both reviewers recommend the acceptance of the paper in the current form. We thank the reviewers for time and effort reading the manuscript.